# Skin Anti-Inflammatory Potential with Reduced Side Effects of Novel Glucocorticoid Receptor Agonists

**DOI:** 10.3390/ijms25010267

**Published:** 2023-12-23

**Authors:** Enrica Flori, Sarah Mosca, Daniela Kovacs, Stefania Briganti, Monica Ottaviani, Arianna Mastrofrancesco, Mauro Truglio, Mauro Picardo

**Affiliations:** 1Laboratory of Cutaneous Physiopathology and Integrated Center of Metabolomics Research, San Gallicano Dermatological Institute, IRCCS, 00144 Rome, Italy; sarah.mosca@ifo.it (S.M.); daniela.kovacs@ifo.it (D.K.); stefania.briganti@ifo.it (S.B.); monica.ottaviani@ifo.it (M.O.); 2Microbiology and Virology, San Gallicano Dermatological Institute, IRCCS, 00144 Rome, Italy; arianna.mastrofrancesco@ifo.it (A.M.); mauro.truglio@ifo.it (M.T.); 3Istituto Dermopatico dell’Immacolata, IDI-IRCCS, 00167 Rome, Italy; m.picardo@idi.it

**Keywords:** skin, glucocorticoids, inflammation, inflammatory disease, skin lipids, keratinocytes, fibroblasts

## Abstract

Glucocorticoids (GCs) are commonly used in the treatment of inflammatory skin diseases, although the balance between therapeutic benefits and side effects is still crucial in clinical practice. One of the major and well-known adverse effects of topical GCs is cutaneous atrophy, which seems to be related to the activation of the glucorticoid receptor (GR) genomic pathway. Dissociating anti-inflammatory activity from atrophogenicity represents an important goal to achieve, in order to avoid side effects on keratinocytes and fibroblasts, known target cells of GC action. To this end, we evaluated the biological activity and safety profile of two novel chemical compounds, DE.303 and KL.202, developed as non-transcriptionally acting GR ligands. In primary keratinocytes, both compounds demonstrated anti-inflammatory properties inhibiting NF-κB activity, downregulating inflammatory cytokine release and interfering with pivotal signaling pathways involved in the inflammatory process. Of note, these beneficial actions were not associated with GC-related atrophic effects: treatments of primary keratinocytes and fibroblasts with DE.303 and KL.202 did not induce, contrarily to dexamethasone—a known potent GC—alterations in extracellular matrix components and lipid synthesis, thus confirming their safety profile. These data provide the basis for evaluating these compounds as effective alternatives to the currently used GCs in managing inflammatory skin diseases.

## 1. Introduction

Glucocorticoids (GCs) are lipophilic steroid hormones produced by adrenal cortical glands in response to stress. They play a crucial role in regulating several physiological and pathological processes, such as inflammation and immunological response. For these reasons, they are among the most powerful and frequently used drugs for topical therapy of inflammatory skin diseases [1,2]. The effects of GCs depend on their binding to the glucocorticoid receptor (GR), a ligand-inducible transcription factor located in an inactive complex with several other proteins in the cytoplasm when unbound. After interacting with GCs, GR undergoes a conformational change and translocates from the cytosol to the nucleus to bind glucocorticoid-receptor-responsive elements (GREs) in the promoter sequences of target genes [3,4,5]. Therefore, the GC–GR interaction leads to the transactivation or transrepression of immunosuppressive and inflammatory genes, respectively [3,4].

Alternatively, GR can indirectly modulate mRNA levels by interacting with other transcription factors in the nucleus, such as AP1, NF-κB, and STATs [5,6]. These GR activities occur slowly, over hours or days, and are known as the genomic mechanisms of GC action. Data in the literature also show non-genomic activities of GCs, not involving receptor translocation into the nucleus, making them more rapid and not requiring gene transcription or protein synthesis [7,8,9]. Non-genomic effects of GCs may also alter multiple downstream signaling pathways, including the proteins p38 MAPK, JNK, PKC, PKA, and ERK1/2, thus having modulatory effects on NF-κB and inflammation [6]. These GC effects can be GR-mediated or involve nonspecific interactions with the cell membrane [9].

Due to their anti-inflammatory and immunosuppressive properties, synthetic GCs represent the mainstay protocol for treating inflammatory skin disorders, such as atopic dermatitis, psoriasis, and hidradenitis suppurativa [3,10,11,12,13,14]. Although GCs can be highly effective in managing pathological conditions, their use may be associated with side effects affecting both the epidermis and dermis. Among these side effects, there have been reports of skin thinning, reduced elasticity, increased permeability, and transepidermal water loss, leading to a dysfunctional skin barrier and promoting subsequent skin atrophy. Skin atrophy is one of the most significant negative effects of acute or chronic topical GC use [3,15]. This downside of GCs’ use is thought to be linked to their genomic transactivation activity [3,16]. The first changes related to skin atrophy occur in the epidermis: the proliferation of keratinocytes is inhibited while their maturation is accelerated [15,17]. Moreover, skin barrier function is impaired through a significant inhibition of the synthesis of epidermal lipids, such as ceramides, triacylglycerols, unsaturated free fatty acids, and cholesterol [17,18]. GC-induced skin atrophy also involves the dermis, with direct inhibition of fibroblast proliferation, depletion of mucopolysaccharides, elastin fibers, matrix metalloproteases (MMPs), and inhibition of collagen synthesis [15,18]. Some evidence showed that dexamethasone (DEX), a known potent steroid GR ligand (full agonist) with atrophogenic effects [19,20], reduces the levels of hyaluronan (HA), a key component of the cutaneous extracellular matrix, by suppressing the activity of hyaluronan synthase 2 (HAS2) [19,21]. DEX has also been shown to dramatically reduce mRNA expression levels of collagen type I and III in human fibroblasts and rat skin, resulting in impaired skin function and wound healing [19,22].

The identification of new classes of GR activators that retain anti-inflammatory therapeutic activity without interfering with the expression of GR target genes involved in side effects is an important research goal. Various attempts have been made to develop novel optimized GR ligands, such as the selective glucocorticoid receptor agonists (SEGRAs), which may exert anti-inflammatory effects with fewer side effects [23,24,25,26]. The non-steroidal compound A has been one of the first selective GR modifiers to be characterized and it is able to repress the NF-κB signaling pathway without GR-mediated genomic activity [23,24,26].

The present study aimed at evaluating the ability of two selected novel chemical entities (NCEs), DE.303 (DE) and KL.202 (KL), specifically designed as non-transcriptional GR ligands to interfere with the activation of crucial signaling pathways involved in the inflammatory process in primary keratinocytes. The effects were compared with those obtained with DEX as a representative of transcriptional GR ligands. In addition, we employed primary cultures of both keratinocytes and fibroblasts to investigate the atrophic potential of the molecules on two cell types representative of the epidermal and dermal compartments, respectively. In contrast to DEX, we found a generalized safety profile of the DE and KL dosage used against the occurrence of skin atrophy. Taken together, our results provide a rationale for further in vivo studies to evaluate the potential efficacy of the two compounds for treating skin inflammatory disorders with fewer side effects.

## 2. Results

### 2.1. DE and KL Did Not Induce GR Transcriptional Activation

DE and KL were designed as novel non-transcriptionally acting GR ligands to avoid side effects linked to GR-mediated transcriptional regulation (chemical structure and molecular docking of DE and KL are shown in Appendix A).

We first exposed NHKs to increasing concentrations (0.01 to 5 µM) of DE and KL in parallel with DEX, a benchmark potent steroid ligand [20]. Cell viability, assessed by MTT assay, demonstrated the absence of cytotoxicity at the doses tested for all three compounds (Appendix A). According to these results and in agreement with previous literature data for DEX [20,27,28,29,30,31], the dose of 1 µM was selected for subsequent experiments. Moreover, this hormone concentration has been shown to saturate the receptor and has a potent transcriptional effect on keratinocytes [32]. Then, we verified the inability of the selected compounds to induce the GR transactivation. On NHKs, we compared the effects of DE and KL with DEX on the GR nuclear translocation and the phosphorylation at Ser211, critical for the transactivation activity of the receptor [33]. The treatment with DEX for 1 h induced a significant nuclear translocation of the GR (Figure 1a), as well as its phosphorylation (Figure 1b), whereas DE and KL, at the same dose, did not induce similar effects. Immunofluorescence analysis confirmed the receptor translocation with clear nuclear staining following DEX treatment in comparison to that observed in control keratinocytes. On the contrary, low nuclear immunoreactivity with no differences compared to untreated cells was detected for DE and KL (Figure 1c). The expression of the Ser211 phosphorylated form of GR increased in response to DEX. Again, the treatment with DE and KL did not result in the same increase in the signal (Figure 1d), indicating that the compounds failed to transactivate the receptor. 

Then, we compared the gene expression profile of NHKs treated with DEX, DE, or KL for 24 h. A total of 93 genes related to GR-mediated transcriptional regulation were evaluated using the gene expression array card system. The expressions of *ANGPTL4*, *BMPER*, *DUSP1*, *ERRFI1*, *FKBP5*, *GLUL*, *KLF9*, *MT2A*, *RGS2*, and *TSC22D3* were significantly upregulated by DEX treatment, whereas *ADARB1*, *ARID5B*, *HAS2*, *POU2F2*, and *RHOJ* transcripts were downregulated. DE and KL were ineffective in inducing similar changes (Figure 1e and Appendix A). The real-time RT-PCR analysis of well-known GR target genes confirms the absence of GR transcriptional modulation by DE and KL (Figure 1f). 

### 2.2. DE and KL Inhibited TNF-α-Induced Inflammatory Cytokines

To evaluate the anti-inflammatory properties of the tested compounds, NHKs were stimulated with TNF-α (20 ng/mL) for 24 h in the presence or absence of the molecules, and the amount of the pro-inflammatory secreted cytokines IL-6 and IL-8 was evaluated. As expected, TNF-α strongly increased the release of IL-6 and IL-8, and the co-treatment with DE and KL significantly reduced this induction comparably to DEX (Figure 2a). We also examined whether GR-mediated signaling was critically involved in the anti-inflammatory activity of the tested molecules. To this aim, NHKs transiently transfected with siRNA for GR (siGR) or control (siCtr) (insert in Figure 2b) were exposed to TNF-α in the presence or absence of the tested molecules, and the levels of the produced pro-inflammatory cytokines were evaluated. The treatment with TNF-α strongly increased the amount of secreted IL-6 and IL-8 in both siGR and siCtr cells. However, this induction was significantly reduced by the co-treatment with the GR ligands only in siCtr cells, indicating that the binding to the GR is necessary to retain the anti-inflammatory activity of the selected GR activators (Figure 2b). 

### 2.3. DE and KL Counteract the Pro-Inflammatory Effects Induced by TNF-α

In addition to the transactivation activity, GR is able to interfere with pro-inflammatory transcription factors, such as NF-κB, AP-1, and JAK/STATs, through direct protein-protein interactions that impair their transcriptional activity [6]. NF-κB is a master regulator of pro-inflammatory mediators such as IL-6 and IL-8 [34,35]. We investigated whether the selected GR ligands could interfere with the activation of the NF-κB pathway. To this end, we analyzed the nuclear translocation of this transcription factor after treatment with TNF-α, in the presence or absence of the compounds. The exposure of NHKs to TNF-α significantly increased the NF-κB expression in the nuclear fractions. This induction was significantly reduced by the co-treatment with the tested GR activators, analogously to DEX (Figure 3a). The immunofluorescence analysis revealed that the presence of the compounds during the stimulation with TNF-α reduced the number of cells with nuclear NF-κB staining, confirming the inhibitory effect on NF-κB activity (Figure 3b). 

To deepen the molecular mechanism underlying the anti-inflammatory activity exerted by the tested GR ligands, we analyzed the ability of the compounds to interfere with the phosphorylation profiles of kinases belonging to crucial signaling pathways involved in the control of the inflammatory process. As expected, NHKs exposed to TNF-α rapidly and significantly increased the protein expression of phospho-STAT1/3, phospho-c-JUN, phospho-AKT, phospho-ERK, and phospho-p38. This induction was significantly reduced by the co-treatment with DE or KL. DEX was used as the positive control (Figure 4a). Then, we moved on to the evaluation of the efficacy of the selected compounds on dampening the pro-inflammatory JAK/STATs signaling, the most crucial pathway implicated in cytokines synthesis [36]. To this aim, we compared DE and KL with the anti-inflammatory drug tofacitinib (TOFA), a well-known Janus kinase 1/3 (JAK1/3) inhibitor [37,38]. NHKs were stimulated with TNF-α for 24 h in the presence or absence of DE, KL, or TOFA, and the amount of the secreted IL-6 and IL-8 was evaluated. Individually, the GR ligands resulted as efficiently as TOFA in counteracting the TNF-α-mediated upregulation of IL-6 and IL-8 release (Figure 4b).

### 2.4. DE and KL Safety Profile in Different Skin Cell Populations

The most prominent irreversible cutaneous adverse effect of extended GCs topical application is skin atrophy, which involves both the epidermis and dermis and is associated with the impairment of the skin barrier through inhibition of lipid synthesis and changes in extracellular matrix (ECM) proteins [15,17,18,19,21,39,40]. To investigate the atrophic potential of DE and KL, we first investigated the mRNA expression level of hyaluronan-metabolizing enzymes (HAS-2 and HAS-3) and MMPs, as key ECM remodeling components, in NHKs treated with the selected compounds. As expected, DEX, which is a strongly atrophogenic GC [19], caused a significant downregulation of all the evaluated genes, whereas DE and KL were not able to induce similar effects, except for *HAS-2*, whose expression was decreased also with KL treatment (Figure 5a). Moreover, we evaluated the fallouts on NHK lipid profile of the selected compounds in comparison to DEX exposure. DEX produced a reduction of cholesterol and fatty acids, in both free and bound form (CH, FFAs, FAMEs), in accordance with reduced lipid synthesis demonstrated for GCs topical application [17]. On the other hand, neither DE nor KL demonstrated impairment of NHK levels of the evaluated lipids, indicating the absence of the atrophogenic effect of these compounds (Figure 5b). 

Then, we evaluated ceramides (CERs), as the lipids mainly involved in the skin barrier functions. CERs are sub-classified based on molecular structure. In keratinocytes, most abundant CER subclasses contain one of two fatty acid moieties, the nonhydroxy [N] and α-hydroxy [A] moieties, bound to one of three sphingoid base moieties: sphingosine [S], dihydrosphingosine [DS], and phytosphingosine [*p*]. We profiled both the α-hydroxy-FA-CER (CER-ADS, CER-AS, and CER-AP) and nonhydroxy-FA-CER (CER-NDS, CER-NS, and CER-NP) subclasses using a scheduled multiple reaction monitoring (MRM) technique (Appendix A). Dex caused a broad reduction of total amount of CERs (Figure 5c) and relative content of each ceramide subclasses (Figure 5d,e) in keratinocytes, leading to a relevant impairment of essential lipids for the maintenance of the skin barrier integrity. Both DE and KL treatment did not result in any ceramide alteration when compared to untreated keratinocytes (Figure 5c–e), confirming the absence of the atrophogenic effect of these molecules. The percentage distribution of the six evaluated ceramide subclasses showed that CER-NS and CER-NDS were the most abundant classes. Moreover, DEX changed the proportions of ceramide classes in the keratinocytes, causing a decreased relative abundance of the class of CER-NS compared to untreated cells, while both DE and KL determined an increased relative amount of CER-NS (Figure 5f).

In the dermis, GCs decrease the proliferation of fibroblasts and the production of ECM proteins [18]. In NHFs, the treatment with DEX resulted in cellular flattening and enlarged shape, as assessed by microscopic analysis followed by quantitative image measurement of cell area (Figure 6a). NHFs exposed to DEX showed the presence of increased and widespread actin stress fibers detected by phalloidin staining (Figure 6b). DEX-treated fibroblasts also revealed a significant reduction in the cell number (Figure 6c), which was reflected in a parallel decrease in the growth rate, evaluated as the number of cells expressing the proliferation marker Ki67 (Figure 6d). Contrarily, NHFs incubated with DE and KL retained their bipolar spindle-like morphology and exhibited no differences in cytoskeleton organization and no slowing of proliferation compared with the control (Figure 6a–d). In accordance with the enlarged shape induced by DEX to fibroblasts, a significant increase in the amount of bound fatty acids, mainly present in the cell membranes, was observed in these cells after DEX exposure. In particular, the treatment with DEX caused a trend of increased production of membrane phospholipids enriched with polyunsaturated fatty acids (PUFA). On the other hand, DE and KL, which did not alter fibroblast morphology, did not produce variation in the quantity of bound fatty acids either (Figure 6e). As for NHKs, only DEX induced a strong decrease in *HAS-3*, *MMP1*, *MMP2*, and *MMP3*. As regards *HAS-2*, both DEX and KL determined a reduction of its transcript. No significant modification of *COL1A1* and *COL3A1* mRNA expression was observed following the different treatments (Figure 6f). 

## 3. Discussion

The identification of more selective and safer GR modulators remains an unsolved challenge. The present findings support the potential benefits of the two tested non-transcriptionally acting GR ligands DE.303 and KL.202. Both chemical entities demonstrated anti-inflammatory capacity by interfering with the activation of pivotal signaling pathways involved in the control of the inflammatory process. Meanwhile, the compounds maintained a generalized safety profile against the occurrence of skin atrophy, one of the most relevant side effects of topical GCs. Subsequent time-dependent studies would define the more active dose for each compound, which was not the aim of these preliminary studies. 

The synthetic GCs were introduced as potent anti-inflammatory drugs for more than 70 years and remain the frontline for the treatment of inflammatory skin diseases, including psoriasis and atopic dermatitis [3,10,11,12,13,41,42]. Unfortunately, although they represent a very effective therapy, they are associated with undesired local effects on the skin and systemic side effects, limiting their long-term utility. Balancing therapeutic benefits and minimizing side effects are pivotal in dermatological clinical practice, and continuous efforts have been made to separate the beneficial anti-inflammatory action from the atrophogenic side effects linked to GR transactivation. The activation and function of GCs are very complex, with genomic and non-genomic effects, and the mechanisms of action responsible for the anti-inflammatory nature of the GR are still far from being fully understood. When the GCs bind to the GR, the latter translocates to the nucleus, where it performs its anti-inflammatory functions both as a monomeric and as a homodimer protein.

The monomer is mainly known for its transrepression action, involving the GR-mediated interference with the transcriptional activity of pro-inflammatory transcription factors, such as NF-κB and AP-1. Instead, the GR homodimer mainly mediates transactivation to directly control the expression of several well-known GREs of anti-inflammatory genes, such as TSD22D3, DUSP1, FKBP5, SGK1, and IkBα [5,6,16,20].

Although important anti-inflammatory genes are transcribed in response to the activities of dimeric GR, this mechanism is linked to metabolic side effects that limit the prolonged administration of GCs in the treatment of inflammatory skin disorders. FAKBP5, for example, is one of the major GR target genes in the skin involved in immune regulation by impairing both NF-κB’s nuclear translocation and transcriptional activity. However, on the other hand, it acts as an atrophogene during chronic skin treatment with steroids [39,43,44]. Moreover, the mutual antagonism between GR and NF-κB has been described to occur also via the GR’s rapid non-genomic-activity, through the ability of the receptor to interfere with multiple downstream signaling pathways, including p38 MAPK, JNK, and ERK1/2, which in turn possess modulatory effects on NF-κB and inflammation [6,45]. In our experiments, the lack of GR nuclear translocation and phosphorylation, as well as the absent induction of expression of well-known GR target genes, confirm the inability of the DE and KL to activate the GR homodimer transactivation. To strengthen the hypothesis of a non-transcriptional anti-inflammatory activity exerted by the tested compounds, we observed a very rapid inhibition of NF-κB activation and a prompt interference with the phosphorylation profiles of kinases belonging to the three well-defined MAPK subfamilies [46]. These kinases play a key role in the host immune response leading to the activation of pro-inflammatory transcription factors, such as NF-κB and AP-1, thus ensuing release of various cytokines, chemokines, and inflammatory mediators. To reduce side effects of long-term therapies with GCs, several studies have investigated the use of JAK inhibitors, such as tofacitinib, for the treatment of several skin inflammatory diseases, with drug approval cases for conditions such as psoriatic arthritis [37,38,47,48]. As for GCs, the use of tofacitinib seems to be linked to side effects such as acne exacerbation, and it is debated whether the therapy with the inhibitor is also associated with the risk of incident venous thromboembolism [49,50,51]. In our experiments, DE and KL were as efficient as tofacitinib in reducing the release of inflammatory mediators, suggesting a possible therapeutic alternative not only to steroid GR ligands such as dexamethasone but also to the JAK inhibitor.

Studies in the literature indicate that not all the described non-genomic GC effects are linked to the receptor, and a distinction between non-genomic specific and nonspecific effects of GC has sometimes been reported [52,53]. The GR silencing experiments added important insights into the mechanism of action of DE and KL, demonstrating that both compounds need to bind to the GR to retain their anti-inflammatory activity. Moreover, the docking results indicated a more viable alternative conformation for DE and KL in terms of GR binding location and energy affinity compared to the DEX reference. This could explain the lack of transactivation and the different effects we observed for DE and KL. The GR is not a rigid molecule. It changes conformation upon binding to a ligand or when it associates with another molecule [16,54]. Each interaction of GR with ligands may affect its conformation, leading to changes in its stimulation, spatial distribution within the cell, and binding to different molecular partners, thus resulting in various effects. We cannot exclude that treatment with both molecules for prolonged times or at higher doses may induce some GR-dependent side effects. However, the fact that DE and KL at 1 µM proved to be as effective as well-documented anti-inflammatory molecules, such as DEX [17,24,25,26,27,28] and tofacitinib [34,35], suggests that this dosage can allow the achievement of therapeutic effects without detectable side effects.

In the last years, research efforts focused on the development of glucocorticoid-like molecules with high efficacy and low side effects. Thus, the concept of SEGRAs rises, with the aim to develop newer GR ligands with a lower side effect profile [20,26,39,55,56]. The non-steroidal compound A (CpdA) has been one of the first SEGRA to be characterized. Although it does not promote GR dimerization and transactivation, it retains anti-inflammatory potential by repressing the NF-κB signaling pathway, without inducing the expression of skin atrophy markers, when topically applied in in vitro and in vivo mouse models [39,56]. However, CpdA lability, in combination with a narrow therapeutic range, requires further studies to consider its therapeutic employment appropriate [26]. The SEGRA LEO134310 demonstrated efficacy after topical application in human and mouse skin models, with significantly less skin atrophy compared to classical steroids. Safety, tolerability, and pharmacodynamics effects of LEO134310 were evaluated in a phase 1b study in adults with chronic plaque psoriasis, but the data have not yet been published [55,57]. 

Skin atrophy is a major unwanted side effect of topical GCs, involving all skin compartments, and is characterized by severe hypoplasia, loss of elasticity, dysfunctional skin barrier, and overall skin thinning. The earliest degenerative changes are found in the epidermis, with reduction of keratinocyte size and of epidermal cell layers, damping of the stratum corneum, and depletion of the lipid content. The dermis is altered by direct inhibition of fibroblast proliferation, depletion of mucopolysaccharides (hyaluronic acid), elastin fibers, MMPs, and inhibition of collagen synthesis, resulting in rearrangement of the geometry of the dermal fibrous network [15]. Atrophogenic changes can be found also in hair follicles, sebaceous glands, or dermal adipose tissue [43,58]. For the development of novel optimized GR ligands to be used for therapeutic purposes, predictive pre-clinical test systems represent an important tool to determine the atrophogenic potential [40,58]. The suppression of HAS2 expression in keratinocytes and fibroblasts constitutes a valid and fast surrogate marker of GC-induced depletion of hyaluronan (HA), one of the most abundant components of the ECM [18,19,21]. According to data in the literature, DEX treatment significantly reduced *HAS2* mRNA levels in both keratinocytes and fibroblasts. Interestingly, DE and KL showed opposing effects on *HAS2* expression, suggesting a safer profile for DE. Contrarily to DEX, both molecules did not affect the expression of MMPs, ECM-degrading enzymes produced by fibroblasts and keratinocytes, which are of critical importance for ECM homeostasis in the skin. Despite a significant suppression of HAS2 and MMPs levels, we did not observe any effect of DEX on *COL1A1* and *COL3A1* mRNA expression in fibroblasts using our 24 h treatment. This result is in agreement with previous reports showing that GC-induced effects on collagen metabolism are delayed and represented an endpoint of GC-induced skin damage, whereas suppression of HAS2 activity depicts an early process [18,19]. 

CERs are a major product of terminal differentiation of keratinocytes and are important for barrier function of SC. Topical GCs are known to reduce skin barrier integrity through their capability to impair the composition of epidermal lipids, such as CERs, CH, and FAs. NHKs induced to differentiate by increased calcium in the growth media effectively reproduce the process of SC generation. DEX caused a broad reduction of CERs as well as CH and FAs in differentiated NHKs. Interestingly, DE and KL maintain the lipid composition underlying the improved safety profiles of these molecules. Consistent with previous reports [58,59], we observed an increase in cell size, a more diffuse actin stress fiber distribution, and a reduction in the growth rate of fibroblasts after the treatment with DEX. Contradictory results have been reported describing how GCs can both promote and inhibit fibroblast proliferation, probably depending on the different experimental conditions employed [60]. Our data are in line with the previous studies describing a reduction in the mitotic index associated with DEX, which, in turn, has been linked to the promotion of the atrophic state of the dermis observed with GC therapy.

The observed morphologic changes are characteristic of senescence phenotype. Senescent cells, in fact, show flattened and enlarged shapes with enhancement of actin stress fibers [61]. Senescence induction is a complex process characterized by high metabolic activity and a series of molecular changes acquired after an initial growth arrest. Lipid metabolism is actively involved in the senescence process, as documented in senescent cells by global alteration in lipid composition, leading to extensive morphological changes through the membrane [62]. During the induction of senescence or aging phenomenon in human primary fibroblasts, lipid synthesis plays a key role through long-chain fatty acid production through the increased activity of fatty acid synthase [63]; cell membrane phospholipid accumulation of phosphatidylcholine molecules with incorporated PUFA, and in particular arachidonic acids [64]; and phosphatidylglycerols and lysophosphatidylglycerols with at least one PUFA [65]. Hence, the absence of changes in cell size, cytoskeletal organization and growth, and lipid amount and composition evidenced after DE and KL is a further indication of a general safety profile of the latter two compounds compared with DEX. However, further experiments would be needed to rule out the possibility that longer treatment periods or higher doses of the compounds might stimulate some effects linked to skin atrophy promotion.

The complexity of GCs’ action makes the development of safer and more effective therapies very difficult. A better understanding of how GR exactly works and how ligands influence the effects of GR may represent an important step toward the discovery of new classes of GCs characterized by reduced side effects. Taken together, our results identify the anti-inflammatory properties in association with limited atrophogenic effects of the two novel GR ligands DE.303 and KL.202, thus providing a rationale for further investigation of their therapeutic efficacy for the treatment of inflammatory skin disorders.

### Limitations of the Study

A limitation of our study is the in vitro experimental approach, which does not consider the interactions between different cell types and the influence of the surrounding microenvironment. In addition, the current experimental model makes it difficult to translate in vitro concentrations into in vivo doses and to simulate the effects of long-term exposure. Further studies on an experimental model more reliable to reproduce in vivo effects, such as 3D-skin equivalents, will help to stabilize the optimal concentrations for efficacy or binding to GR for each compound.

## 4. Materials and Methods

### 4.1. Materials

M154 defined medium, human keratinocyte growth supplements (HKGS), fetal bovine serum (FBS), L-glutamine, penicillin/streptomycin, trypsin/EDTA, and D-PBS were purchased from Invitrogen Technologies (Monza, Italy). DMEM basal medium was purchased from Euroclone (Milan, Italy). Sebomed Basal Medium (Cat#F8205) was purchased from Merck (Darmstadt, Germany). Recombinant human tumor necrosis factor-α (TNF-α) was from R&D System (Minneapolis, MN, USA). AurumTM Total RNA Mini kit, SYBR Green PCR Master Mix, and Bradford reagent were from Bio-Rad (Milan, Italy). PrimeScriptTM RT Master Mix (Perfect Real Time) (#RR036A) for cDNA synthesis was from Takara Bio Inc. (Kusatsu, Shiga, Japan). IL-6 and IL-8 ELISA kits were from Diaclone SAS (Besancon Cedex, France). NE-PER nuclear and cytoplasmic extraction reagents were from Thermo Fisher Scientific (Monza, Italy). β-Actin antibody (A5441) (1:10,000) and GAPDH antibody (G9545) (1:5000) were from Sigma-Aldrich (Milan, Italy). The antibodies for glucocorticoid receptor (D6H2L) (#12041) (1:1000), phospho-glucocorticoid receptor (Ser211) (#4161), phospho-p38 MAP kinase (Thr180/Tyr182) (#4511) (1:1000), p38 MAPK (#9212) (1:1000), phospho-AKT (Ser473) (#4060) (1:1000), AKT (#2920) (1:1000), Histone H3 (96C10) (#3638) (1:1000), secondary anti-mouse IgG HRP-conjugated antibody (#7076) (1:3000), and anti-rabbit IgG HRP-conjugated antibody (#7074) (1:8000) were purchased from Cell Signaling (Danvers, MA, USA); the anti-NF-κB p65 antibody [E379] (ab32536) (1:3000) was purchased from Abcam (Cambridge, UK). Amersharm ECL Western Blotting Detection Reagent was from GE Healthcare (Buckinghamshine, UK). RIPA lysis buffer, broad-spectrum protease inhibitor cocktail, and broad-spectrum phosphatase inhibitor cocktail were from Boster Biological Technology Co. (Pleasanton, CA, USA). Non-specific siRNA (sc-37007) and GR siRNA (sc-35505) were from Santa Cruz Biotechnology (Dallas, TX, USA). The Amaxa^®^ human keratinocyte Nucleofector kit was from Lonza (Basel, Switzerland). 

### 4.2. New Chemical Ligands 

The NCEs DE.303 (DE) and KL.202 (KL) were designed (in in silico screening) as non-transcriptionally acting GR ligands and synthesized by PPM Services SA-A Nogra Group Company to inhibit inflammatory processes avoiding the side effects linked to GR-mediated transcriptional regulation. The non-steroidal compounds were virtually docked into the binding pocket of the GR and assigned a score that reflected their predicted affinity of binding. The molecular docking webserver SwissDock [66] was used to predict the molecular interactions that may occur between the target receptor GR and the NCEs. The results were analyzed with UCSF Chimera viewer [67], using the known ligand DEX as a reference. For the GR, a model ligated with DEX was available (4UDC), so it was just necessary to remove the ligand and obtain a reliable apo structure for the docking procedure. The first step was to dock DEX to the receptor in order to obtain a reference framework regarding position, energy, and fitness for a known ligand, trying all possible interactions, without restricting to any binding pocket location. The conformation highlighted in yellow produced the best energy values, had no clashes, and occupied the same space in the binding pocket as the original ligand for the holo 4UDC structure (Appendix A). Once we set this conformation as a reference, we proceeded with docking DE and KL using the same method and parameters. It is important to note that the overall most viable docking solutions in terms of energy (highlighted in yellow), for both molecules, lie outside the receptor, not in the DEX binding pocket. The molecules docked in the DEX pocket were ranked in a lower position (highlighted in orange), but they showed good affinity nonetheless. The best final conformations of DE and KL, and the ligand-protein binding energies, are shown in Appendix A, respectively.

### 4.3. Cell Cultures

Primary cultures of normal human keratinocytes (NHKs) and dermal fibroblasts (NHFs) were isolated from the neonatal foreskin following a previously described procedure [68,69]. NHKs were maintained at 37 °C under 5% CO_2_ in the defined medium M154 with HKGS, L-glutamine (2 mM), penicillin (100 u/mL), and streptomycin (100 μg/mL). NHKs were sub-cultivated once a week, and the experiments were carried out in cells between passages 2 and 4. NHFs were maintained at 37 °C under 5% CO_2_ in DMEM supplemented with 10% FBS, L-glutamine (2 mM), penicillin (100 u/mL), and streptomycin (100 μg/mL) and used between passages 2 and 10. Immortalized human SZ95 sebocytes [70], showing morphologic, phenotypic, and functional characteristics of normal human sebocytes, were cultured in Sebomed^®^ basal medium, supplemented with 10% FBS, L-glutamine (2 mM), penicillin (100 u/mL), streptomycin (100 μg/mL), recombinant human epidermal growth factor (5 ng/mL), and CaCl_2_ (1 mM) in a humidified atmosphere containing 5% CO_2_ at 37 °C. Cell lines were routinely tested for mycoplasma detection. The Institute’s Research Ethics Committee (IFO) approval was obtained to collect samples of human material for research (Prot CE/286/06, approved on 21 April 2006). The study was conducted according to the Declaration of Helsinki Principles. Human samples (primary cell cultures) were collected from patients who had provided written informed consent. For each experiment, at least three different donors were used. Cells were plated and 24 h later were starved to be stimulated with chemicals in fresh medium after 24 h, according to the experimental design.

### 4.4. MTT Assay

After starvation for 24 h, cells treated with DEX, DE, or KL in a 0.01–5 µM dose range for 72 h were then incubated with 3-(4,5-dimethyl-2-thiazolyl)-2,5-diphenyl-2H-tetrazolium bromide (MTT) (1 mg/mL) for 2 hours at 37 °C and lysed in dimethyl sulfoxide (DMSO). The absorbance at 570 nm was measured by a spectrophotometer DTX880 Multimode Detector. The measurement was performed in triplicate for each sample. 

### 4.5. RNA Extraction, Gene Expression Array Card Analysis, and Real-Time RT-PCR

Total RNA was isolated using the AurumTM Total RNA Mini kit, according to the manufacturer’s instructions. Total RNA samples were stored at −80 °C until use. Following DNAse I treatment, cDNA was synthesized using PrimeScriptTM RT Master Mix for the cDNA synthesis kit according to the manufacturer’s instructions. cDNA was loaded on 384-well microfluidic cards designed to perform probe-based TaqMan real-time PCR on an Applied Biosystems^®^ (Milan, Italy) QuantStudioTM 7 Flex instrument (Thermo Fisher Scientific, Monza, Italy). Cards were configured with selected primers and probe sets to analyze 93 target genes and 3 housekeeping genes (18S rRNA, GAPDH, and β-actin). Results were evaluated using QuantStudio™ Design and Analysis Software v2 (Thermo Fisher Scientific). Real-time RT-PCR was performed in a total volume of 10 μL with SYBR Green PCR Master Mix and 200 nM concentration of each primer. Sequences of all primers used are shown in Appendix A. Reactions were carried out in triplicate using a CFx96 Real-Time System (Bio-Rad, Hercules, CA, USA). Melt curve analysis was performed to confirm the specificity of the amplified products. Expression of mRNA (relative) was normalized to the expression of GAPDH mRNA by the change in the Δ cycle threshold (ΔCt) method and calculated based on 2^−ΔCt^.

### 4.6. Western Blot Analysis

Cells were lysed in RIPA lysis buffer supplemented with a protease/phosphatase inhibitor cocktail and then sonicated. Total cell lysates were clarified by centrifugation at 12,000 rpm for 10 min at 4 °C and then stored at −80 °C until analysis. Following spectrophotometric protein measurement, equal amounts of protein were resolved on acrylamide SDS-PAGE and transferred onto nitrocellulose membrane (Amersham Biosciences, Milan, Italy). Protein transfer efficiency was checked with Ponceau S staining (Sigma-Aldrich, St Louis, MO, USA). Membranes were first washed with water, blocked with EveryBlot Blocking Buffer (Bio-Rad Laboratories Srl, Milan, Italy) for 10 min at room temperature, and then treated overnight at 4 °C with primary antibodies (according to data sheet instructions). A secondary anti-mouse IgG HRP-conjugated antibody or anti-rabbit IgG HRP-conjugated antibody was used. Antibody complexes were visualized using enhanced chemiluminescence (ECL) substrate. A subsequent hybridization with anti-β-actin or anti-GAPDH or anti-H3 was used as a loading control. The cytoplasm and nuclear fractions were used to determine the level of NF-κB using NE-PER nuclear and cytoplasmic extraction reagents and anti-NF-κB p65 antibody. The control value was taken as onefold in each case. Protein levels were quantified by measuring the optical densities of specific bands using the UVITEC Mini HD9 acquisition system (Alliance UVItec Ltd., Cambridge, UK). The control value was taken as onefold in each case.

### 4.7. Protein Determination by Sandwich Enzyme-Linked Immunosorbent Assay (ELISA)

Culture supernatants were collected and centrifuged to remove cell detritus. Aliquots were stored at −80 °C until use. IL-6 and IL-8 protein levels were determined using commercially available ELISA kits, according to the manufacturer’s instructions. The measurement was performed in duplicate for each sample. The absorbance at 450 nm was recorded using a DTX880 Multimode Detector spectrophotometer (Beckman Coulter Srl, Milan, Italy). The results were normalized for the number of cells contained in each sample.

### 4.8. Cell Number Analysis

Cell number was evaluated by cell counting using the phase-contrast microscope Axiovert 40C (Zeiss, Milan, Italy). None of the employed chemicals determined positivity to the Trypan Blue exclusion assay test. 

### 4.9. Cell Size Measurement

The measurement of the cell size was performed using the Zen 2.6 (blue edition) software (Zeiss), and the results are expressed as fold change of the mean value/cell ± SD relative to control cell value, which was set as one by definition. At least 200 cells were evaluated for each condition. 

### 4.10. RNA Interference Experiments 

For the RNA interference experiments, NHKs were transfected with 100 pmol siRNA specific for GR (sc-35505; Santa Cruz Biotechnology). An equivalent amount of non-specific siRNA (sc-37007; Santa Cruz Biotechnology) was used as a negative control. Cells were transfected using an Amaxa^®^ human keratinocyte nucleofector kit (Lonza, Basel, Switzerland), according to the manufacturer’s instructions. To ensure identical siRNA efficiency among the plates, cells were transfected together in a single cuvette and plated immediately after nucleofection. 

### 4.11. Immunofluorescence Analysis

NHKs were fixed with 4% paraformaldehyde followed by 0.1% Triton X-100 to allow permeabilization. Cells were then incubated with the following primary antibodies: glucocorticoid receptor (D6H2L) (1:200) (Cell Signaling), phospho-glucocorticoid receptor (Ser211) (1:2000) (Cell Signaling), anti-NF-κB p65 antibody [E379] (1:200) (Abcam), and anti-Ki67 polyclonal antibody (1:400) (Abcam). The primary antibodies were visualized using goat anti-rabbit Alexa Fluor 546 conjugate antibody (1:800) (Thermo Fisher Scientific, Monza, Italy). For staining of actin cytoskeleton, cells were incubated with TRITC-phalloidin (1:800) (Sigma Chemicals). Coverslips were mounted using ProLong Gold Antifade reagent with 4′,6-diamidino-2-phenylindole (DAPI; Invitrogen). The fluorescence signals were analyzed by recording stained images using a charge-coupled device camera (Zeiss, Oberkochen, Germany). The percentage of Ki67-positive cells was analyzed by counting at least 300 cells for each condition, and the results are expressed as % of positive cells/total cells. 

### 4.12. Lipid Extraction

After differentiation induction by maintaining high calcium concentration (1.8 mM) in the culture medium for 5 days, keratinocytes (NHKs) were starved and treated with DEX, DE, or KL (1 µM) for 24 h, always in the presence of calcium 1.8 mM. Lipid extraction was performed according to the Bligh and Dryer procedure with slight modifications [71]. Briefly, lipids were extracted with chloroform/methanol (2:1) after the addition of butylhydroxytoluene (BHT) to prevent oxidation of oxygen sensitive compounds. A mixture of deuterated standards (d6CH; d17C16:0; d98TG48:0; d31CER) were added to control the analytical performance and to calculate the relative abundance of the lipid species detected. The organic layers were collected and evaporated under nitrogen. The dried lipid extracts were stored at −80 °C until the analysis. 

### 4.13. GC-MS Analysis of FAMEs 

Bound fatty acids (FA) were analyzed as FA methyl esters (FAME) obtained after the derivatization reaction described below. For the simultaneous saponification and methylation of bound FA, 250 μL KOH solution (0.5 M) in anhydrous methanol was added to the dried extract and it was incubated at 37 °C for 20 min under constant shaking. A total of 0.5 mL HCl (0.25 M) was added to neutralize the alkaline reaction mixture. After vortex mixing, 0.25 mL K_2_SO_4_ (6.7%) and 1 mL hexane/isopropanol (3:2 *v*/*v*) containing 0.0025% BHT were added, and the mixture was vortexed. After centrifugation, the lipid-enriched upper phase was transferred to an Eppendorf tube and evaporated under nitrogen. The dried FAME extract was dissolved in 20 µL isopropanol (IPA) and was analyzed by gas chromatography–mass spectrometry (GC-MS) to establish FA profiles in the lipid extracts (GC 7890A coupled to an MS 5975 VL analyzer, Agilent Technologies, Santa Clara, CA, USA). The chromatographic separation was carried out on a HP-FFAP (crosslinked FFAP, Agilent Technologies, Santa Clara, CA, USA) capillary column (length 50 m, film thickness 0.52 µm). Helium was used as the carrier gas. The initial GC oven temperature was 40 °C and was linearly ramped up to 240 °C at 8 °C/min. The total run time was 60 min. The injector and the GC-MS transfer lines were kept at 230 °C and 250 °C, respectively. Total ion chromatograms (TIC) were acquired, and areas of single peaks, corresponding to the FAMEs, were integrated with the Agilent MassHunter Workstation Software vB.07.00. The identity of the detected FAME was verified by comparison with authentic standards and matched with library spectral data. The nmol amounts of FAMEs were calculated against the nmol of d31C16:0 generated from d98TG 48:0 [72]. 

### 4.14. GC-MS Analysis of FFAs and CH

Free fatty acids (FFAs) and cholesterol (CH) were analyzed after direct silylation with N,O-bis-(trimethylsilyl)-trifluoroacetamide containing 1% trimethylchlorosilane as catalyst (Sigma-Aldrich, Milan, Italy). After 30 min at 60 °C, the samples were analyzed with GC 7890A coupled to an MS 5975 VL analyzer (Agilent Technologies, Santa Clara, CA, USA). The chromatographic separation was performed on an HP-5MS (Agilent Technologies, Santa Clara, CA, USA) capillary column (30 m × 250 µm × 0.25 µm), using helium as the carrier gas. An oven temperature gradient from 80 °C to 200 °C at 8 °C/min and then to 250 °C at 10 °C/min was used. The injector and the GC-MS transfer lines were kept at 260 °C and 280 °C, respectively. Samples were analyzed in scan mode utilizing electron impact (EI) mass spectrometry. The identity of the detected FFAs and CH was verified by the comparison with authentic standards and the match with library spectral data. The area of the peaks corresponding to the different analytes were integrated with the qualitative analysis software. The nmol amount of FFAs and CH were calculated against the added nmol of d6CH and d17C16:0, respectively.

### 4.15. LC-MS Analysis of Ceramides

CERs analyses were performed using ultraperformance liquid chromatography (1200 HPLC Liquid Chromatography System; Agilent Technologies, Palo alto, CA, USA) coupled to a triple quadrupole mass spectrometer with electrospray ionization (6400 triple quadrupole LC/MS, Agilent Technologies, Palo alto, CA, USA). CERs were separated using a C8 column (Zorbax SB-C8 Rapid resolution HT, 1.8 μm, 2.1 mm × 100 mm; Agilent Technologies, Palo Alto, CA, USA) and a gradient of solvent A (water with 0.1% formic acid) and solvent B (methanol with 0.1% formic acid) as follows: 0.2 mL/min 30% B (0–1 min), 0.2 mL/min 30–70% B (1–2.5 min), 0.2 mL/min 70–80% B (2.5–4 min), 0.3 mL/min 80–90% B 4–8 min), 0.3 mL/min 90%B (8–50 min), 0.2 mL/min 90% B (50–52 min), 0.2 mL/min 90–30% B 52–60 min. The column temperature was maintained at 40 °C. CERs were fragmented using helium as a collision gas and monitored in the positive ion mode by multiple reaction monitoring (MRM) (parameters given in Appendix A). Relative quantification of analytes was performed using the class-specific internal standard d31CER16:0, and results are reported as nmol per million of cells.

### 4.16. Statistical Analysis

Data were represented as mean ± SD of three independent experiments using at least three different donors. Statistical significance was assessed using paired Student’s *t*-test or ANOVA followed by Tukey’s multiple comparisons test using GraphPad Prism software v8.0.2 (Boston, MA, USA). The minimal level of significance was *p* < 0.05.

## Figures and Tables

**Figure 1 ijms-25-00267-f001:**
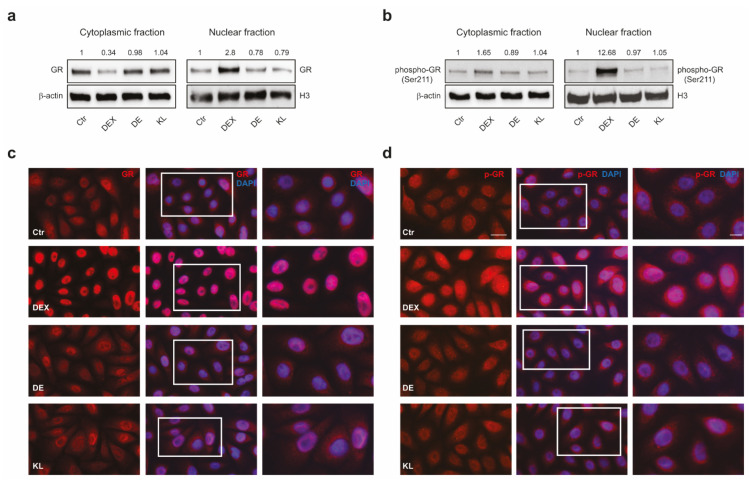
DE and KL did not induce GR transcriptional activation. Western blot analysis and corresponding densitometric analysis of (**a**) GR and (**b**) phospho-GR protein expression at cytoplasmic and nuclear levels on NHKs treated with DEX, DE, and KL (1 µM) for 1 h. β-Actin was used as the loading control in the cytoplasmic fraction, and H3 was used for the nuclear one. Representative blots are shown. Results are expressed as the fold change respect to untreated control cells. (**c**,**d**) Immunofluorescence analysis of (**c**) GR and (**d**) phospho-GR expression in NHKs treated with DEX, DE, and KL (1 µM). Nuclei were counterstained with DAPI. The enlarged view of the selected cells in the white-line-framed areas is shown. Scale bar: 20 µm; enlarged view of the white line-framed areas: 10 µm. (**e**) Array card heatmap illustrating gene expression analysis in NHKs treated with DEX, DE, and KL (1 µM) for 24 h. Red and blue shadings represent higher and lower relative log2 fold change expression levels, respectively. (**f**) Real-time RT-PCR analysis of *ANGPTL4*, *DUSP1*, *ERRF1*, *FKBP5*, *GLUL*, *HAS2*, *MT2A*, *RGS2*, and *TSC22D3* in NHKs treated with DEX, DE, and KL (1 µM) for 24 h. All mRNA values were normalized against the expression of *GAPDH* and were expressed relative to untreated control cells. The data in the graphs are mean ± SD of three independent experiments (* *p* < 0.05 vs. untreated control).

**Figure 2 ijms-25-00267-f002:**
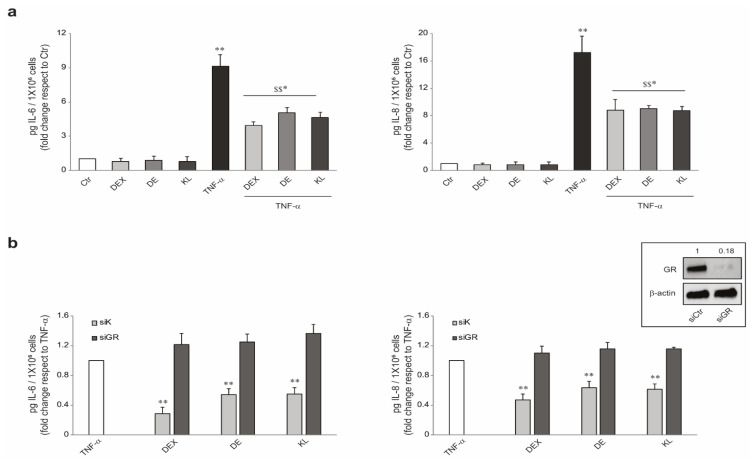
DE and KL counteracted the TNF-α-induced expression of pro-inflammatory cytokines in NHKs. (**a**) IL-6 and IL-8 quantitation by ELISA in NHKs pre-treated with DEX, DE, and KL (1 µM) for 1 h and then stimulated with TNF-α (20 ng/mL) for 24 h. The data are expressed as mean ± SD of three independent experiments (* *p* < 0.05, ** *p* < 0.01 vs. untreated control; $$ *p* < 0.05 vs. TNF-α-stimulated cells). (**b**) IL-6 and IL-8 quantitation by ELISA in NHKs transfected with specific siRNA for the GR gene (siGR) or non-specific siRNA (siCtr) for 24 h, then pre-treated with DEX, DE, and KL (1 µM) for 1 h and stimulated with TNF-α (20 ng/mL) for 24 h. GR protein expression level was evaluated by Western blot. The data are expressed as mean ± SD of three independent experiments (** *p* < 0.01 vs. TNF-α-stimulated cells).

**Figure 3 ijms-25-00267-f003:**
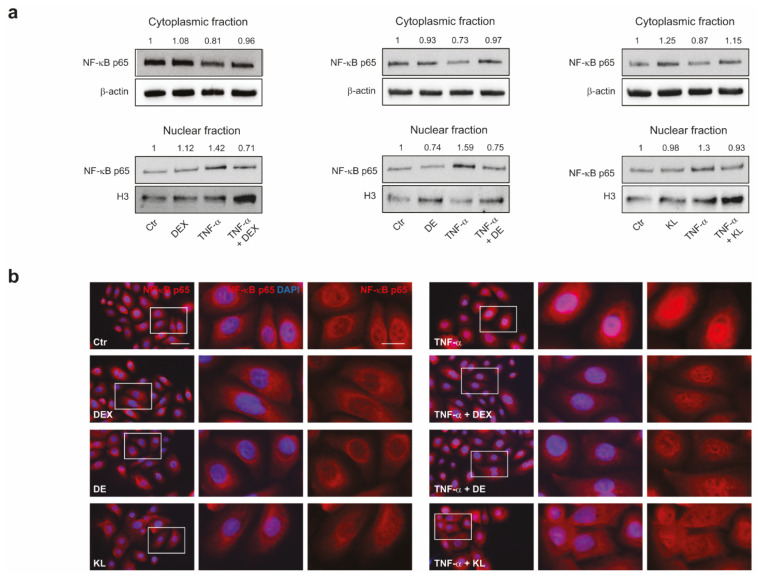
DE and KL counteracted the NF-κB activation induced by TNF-α in NHKs. (**a**) Western blot analysis and corresponding densitometric analysis of NF-κB protein expression at cytoplasmic and nuclear level on NHKs pre-treated with DEX, DE, and KL (1 µM) for 1 h and then stimulated with TNF-α (20 ng/mL) for 1 h. β-Actin was used as the loading control in the cytoplasmic fraction, and H3 was used for the nuclear one. Results are expressed as the fold change respect to untreated control cells. (**b**) Representative immunofluorescence images of NK-kB immunoreactivity in NHKs treated with DEX, DE, and KL and stimulated with TNF-α (20 ng/mL) alone or pre-treated with the compounds (1 µM) for 1 h. The enlarged view of the selected cells in the white line-framed areas is shown. Nuclei were counterstained with DAPI. Scale bar: 50 µm; enlarged view of the white line-framed areas: 20 µm.

**Figure 4 ijms-25-00267-f004:**
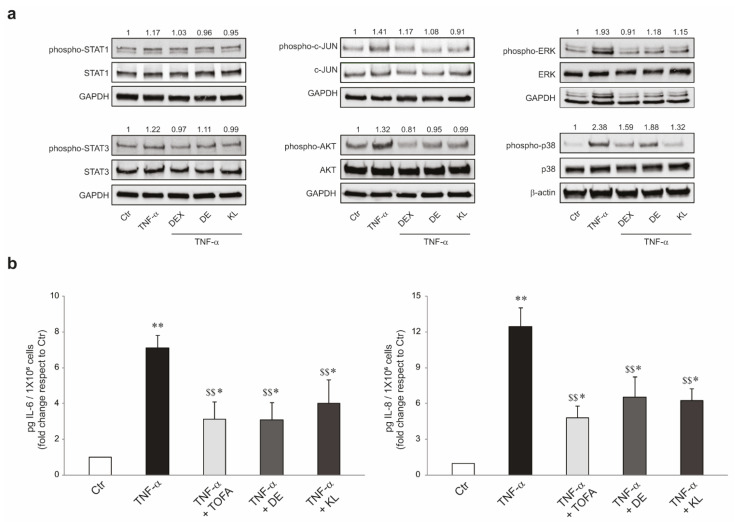
DE and KL counteracted the inflammatory pathways induced by TNF-α in NHKs. (**a**) Western blot analysis and corresponding densitometric analysis of phospho-STAT1, STAT1, phospho-STAT3, STAT3, phospho-c-Jun, c-Jun, phospho-ERK, ERK, phospho-p38, and p38 protein expression in NHKs pre-treated with DEX, DE, and KL (1 µM) for 1 h and then stimulated with TNF-α (20 ng/mL) for 10 min. GAPDH and β-actin were used as endogenous loading control. Results are expressed as the fold change respect to untreated control cells. (**b**) IL-6 and IL-8 quantitation by ELISA in NHKs pre-treated with TOFA (2 µM), DE (1 µM), and KL (1 µM) for 1 h and then stimulated with TNF-α (20 ng/mL) for 24 h. The data are expressed as mean ± SD of three independent experiments (* *p* < 0.05, ** *p* < 0.01 vs. untreated control; $$ *p* < 0.05 vs. TNF-α stimulated cells).

**Figure 5 ijms-25-00267-f005:**
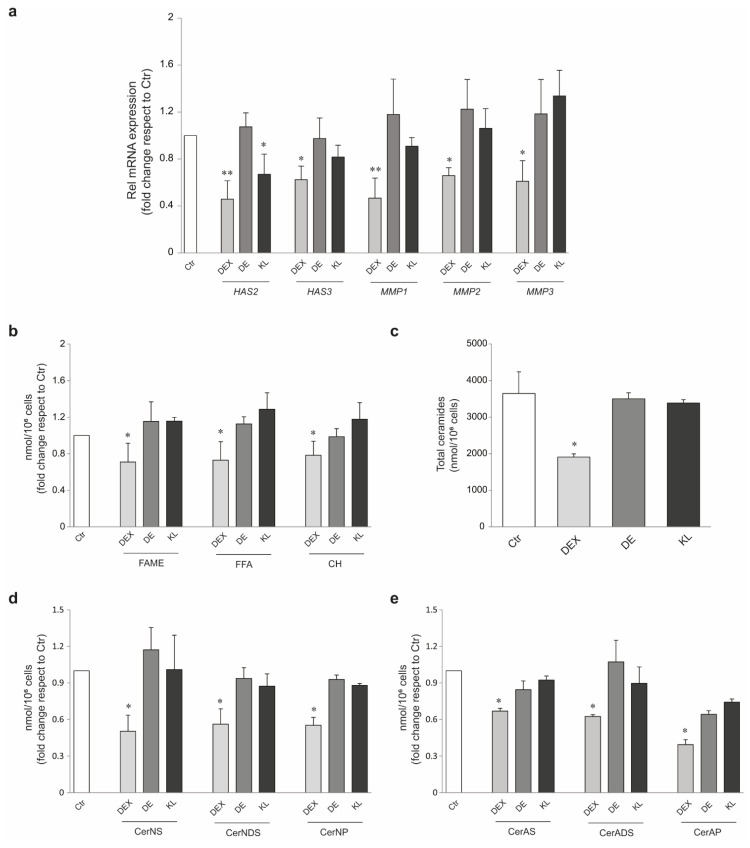
DE and KL safety profile in NHKs. (**a**) Real-time RT-PCR analysis of *HAS2*, *HAS3*, *MMP1*, *MMP2*, and *MMP3* in NHKs treated with DEX, DE, and KL (1 µM) for 24 h. All mRNA values were normalized against the expression of *GAPDH* and were expressed relative to untreated control cells. The data in the graphs are mean ± SD of three independent experiments (* *p* < 0.05, ** *p* < 0.01 vs. untreated control). (**b**) GCMS analysis of FAMEs, FFAs, and CH in differentiated NHKs treated with DEX, DE, and KL (1 µM) for 24 h. Lipid amounts were expressed as fold change with respect to the untreated control. The data in the graph are mean ± SD of three independent experiments (* *p* < 0.05 vs. untreated control). (**c**) LCMS analysis of total amount of CERs in differentiated NHKs treated with DEX, DE, and KL (1 µM) for 24 h. CERa amounts were expressed as nmol normalized to millions of cells. The data in the graph are mean ± SD of three independent experiments (* *p* < 0.05 vs. untreated control). (**d**) LCMS analysis of relative content of non-hydroxy fatty acids (N) ceramide classes in differentiated NHKs treated with DEX, DE, and KL (1 µM) for 24 h. Each ceramide class was reported as fold change with respect to untreated control. The data in the graph are mean ± SD of three independent experiments (* *p* < 0.05 vs. untreated control). Sphingosine (S), dihydrosphingosine (DS), and phytosphingosine (P). (**e**) LCMS analysis of relative content of α-hydroxy fatty acid (A) ceramide classes in differentiated NHKs treated with DEX, DE, and KL (1 µM) for 24 h. Each ceramide class was reported as fold change with respect to untreated control. The data in the graph are mean ± SD of three independent experiments (* *p* < 0.05 vs. untreated control). Sphingosine (S), dihydrosphingosine (DS), and phytosphingosine (P). (**f**) The proportion of ceramide families in differentiated NHKs treated with DEX, DE, and KL (1 µM) for 24 h. Non-hydroxy fatty acids (N), α-hydroxy fatty acids (A), sphingosine (S), dihydrosphingosine (DS), and phytosphingosine (P).

**Figure 6 ijms-25-00267-f006:**
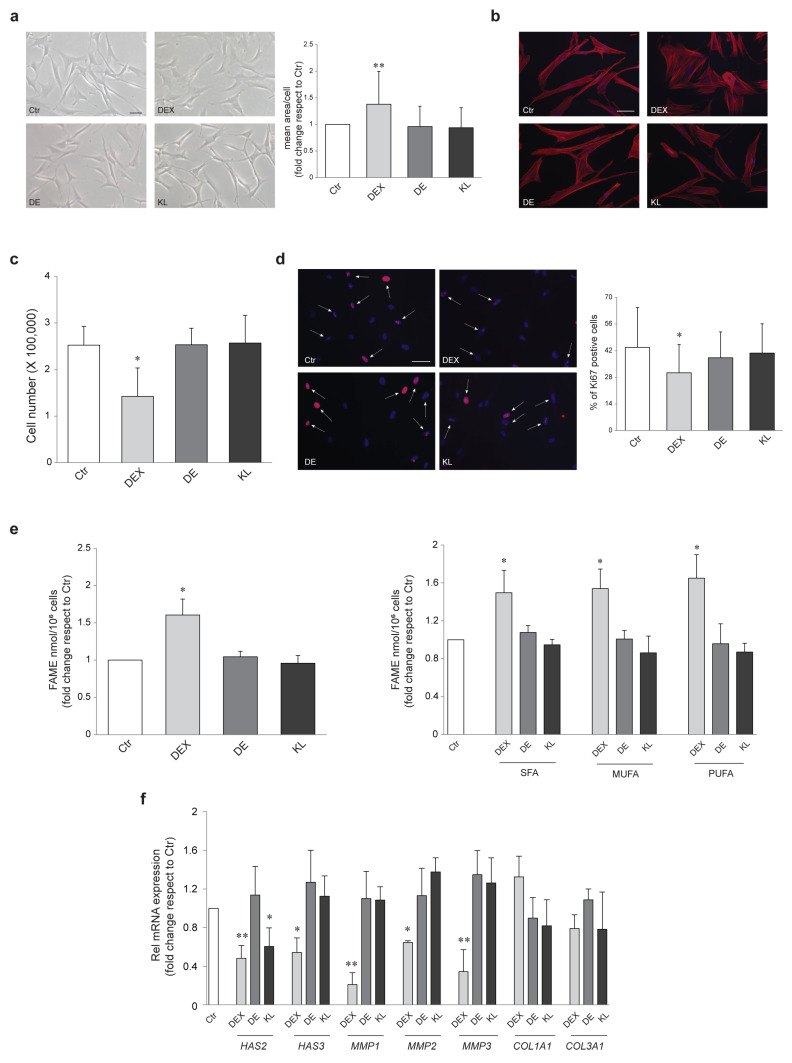
DE and KL safety profile in NHFs. (**a**) Morphological analysis and measurement of cell area of NHFs cells treated with DEX, DE, and KL (1 µM) for 72 h. At least 200 cells were evaluated for each condition (** *p* < 0.01 vs. untreated control). Scale bar: 100 µm. (**b**) TRITC-phalloidin staining on NHFs upon treatment with DEX, DE, and KL. Nuclei are counterstained with DAPI. Scale bar: 50 µm (**c**) Graphs illustrating cell count assay data of NHFs cells treated as above. The data are mean ± SD of three independent experiments (* *p* < 0.05 vs. untreated control). (**d**) Immunofluorescence and quantitative analysis of Ki67-positive NHFs treated with DEX, DE, and KL. Arrows point at cells positively stained for the proliferation marker. Nuclei were counterstained with DAPI (* *p* < 0.05 vs. untreated control). Scale bar: 50 µm. (**e**) GCMS analysis of FAMEs in NHFs treated with DEX, DE, and KL (1 µM) for 72 h. Lipid amounts were expressed as fold change with respect to untreated control (* *p* < 0.05 vs. untreated control). Saturated fatty acids (SFA), monounsaturated fatty acids (MUFA), and polyunsaturated fatty acids (PUFA). (**f**) Real-time RT-PCR analysis of *HAS2*, *HAS3*, *MMP1*, *MMP2*, *MMP3*, *COL1A1*, and *COL3A1* in NHFs treated with DEX, DE, and KL (1 µM) for 24 h. All mRNA values were normalized against the expression of *GAPDH* and were expressed relative to untreated control cells. The data in the graphs are mean ± SD of three independent experiments (* *p* < 0.05, ** *p* < 0.01 vs. untreated control).

## Data Availability

The data presented in this study are available on request from the corresponding author.

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
