# Peer review of "Skin Anti-Inflammatory Potential with Reduced Side Effects of Novel Glucocorticoid Receptor Agonists"

_ijms, 2023, doi:10.3390/ijms25010267_

Round 1

Reviewer 1 Report

Comments and Suggestions for Authors

The study conducted by the authors is really very interesting and well written 

The materials and methods section is well detailed , the discussion part is well explained, figures and tables are very good

I only have minor revisions 

1) in the introduction you need to be clearer about which inflammatory skin diseases glucocorticoids are involved, example psoriasis hs or atopic dermatitis, I leave some references for the authors 

- Martora, F., Martora, L., Fabbrocini, G., & Marasca, C. (2022). A Case of Pemphigus Vulgaris and Hidradenitis Suppurativa: May Systemic Steroids Be Considered in the Standard Management of Hidradenitis Suppurativa?. Skin appendage disorders, 8(3), 265-268. https://doi.org/10.1159/000521712

- Sevilla, L. M., & Pérez, P. (2019). Glucocorticoids and Glucocorticoid-Induced-Leucine-Zipper (GILZ) in Psoriasis. Frontiers in immunology, 10, 2220. https://doi.org/10.3389/fimmu.2019.02220

- DOI: 10.1111/jdv.14891

2) you have to add study limitations section. 

I have no other reviews , otherwise the article is really good 

Author Response

Reviewer 1

The study conducted by the authors is really very interesting and well written. The materials and methods section is well detailed, the discussion part is well explained, figures and tables are very good. I only have minor revisions:

1) in the introduction you need to be clearer about which inflammatory skin diseases glucocorticoids are involved, example psoriasis hs or atopic dermatitis, I leave some references for the authors: Martora, F., Martora, L., Fabbrocini, G., & Marasca, C. (2022). A Case of Pemphigus Vulgaris and Hidradenitis Suppurativa: May Systemic Steroids Be Considered in the Standard Management of Hidradenitis Suppurativa?. Skin appendage disorders, 8(3), 265-268;  Sevilla, L. M., & Pérez, P. (2019). Glucocorticoids and Glucocorticoid-Induced-Leucine-Zipper (GILZ) in Psoriasis. Frontiers in immunology, 10, 2220; DOI: 10.1111/jdv.14891.

We thank the Reviewer for this consideration. In the revised version of the paper, we have inserted the information in the Introduction section (page 2) and we have provided the literature references (page 20).

2) you have to add study limitations section.

As required, we have added the Study limitations section in the text (page 19)

Reviewer 2 Report

Comments and Suggestions for Authors

This study is considered to be a significant research in that the newly developed DE and KL compounds show anti-inflammatory effects, but suppress GR-mediated side effects. However, supplementary explanation is needed for the following points:

1) In Fig. 2, DE and KL do not show anti-inflammatory effects in the absence of GR. Does this experimental result mean that the GR dependent mechanism is essential for the effectiveness of DE and KL? If so, it seems that some GR-dependent side effects are bound to appear, so it would be good if the author's reflections on this appear.

2) In the manuscript, there is no explanation for Figure 6f.

3) The author conducted the experiment at 1 uM considering toxicity data, but I believe that the optimal efficacy concentration or optimal concentration for binding to GR may be different for each compound. Therefore, concentration-dependent experimental results may be important in efficacy experiments such as anti-inflammatory or cell proliferation, but are omitted in this study. In particular, in cell proliferation experiments, it is important to compare the differences in cell proliferation rates of each compound according to concentration and time rather than comparing a single concentration and a single time point. In Figure 6, the possibility that DE and KL also reduce cell numbers at high concentrations cannot be ruled out.

Author Response

This study is considered to be a significant research in that the newly developed DE and KL compounds show anti-inflammatory effects, but suppress GR-mediated side effects. However, supplementary explanation is needed for the following points:

1) In Fig. 2, DE and KL do not show anti-inflammatory effects in the absence of GR. Does this experimental result mean that the GR dependent mechanism is essential for the effectiveness of DE and KL? If so, it seems that some GR-dependent side effects are bound to appear, so it would be good if the author's reflections on this appear.

We thank the Reviewer for this consideration. The GR silencing experiments demonstrated that both compounds need to bind to the GR to retain their anti-inflammatory activity. However, we cannot exclude that treatment with both molecules for prolonged times may induce some GR-dependent side effects. We have added a reflection in the Discussion section (page 13). Furthermore, we have inserted a Study limitations section in the text to highlight the limitations of our experimental system (page 19).

2) In the manuscript, there is no explanation for Figure 6f.

We apologize for the typo in the Figure number (Figure 6e instead of Figure 6f). The results related to Figure 6f are correctly reported at the end of the paragraph 2.4 of the Results section: “As for NHKs, only DEX induced a strong decrease of HAS-3, MMP1, MMP2 and MMP3. As regards HAS-2, both DEX and KL determined a reduction of its transcript. No significant modification of COL1A1 and COL3A1 mRNA expression was observed following the different treatments”.  In the revised version of the manuscript, we have corrected the number of the Figure (page 10).

3) The author conducted the experiment at 1 µM considering toxicity data, but I believe that the optimal efficacy concentration or optimal concentration for binding to GR may be different for each compound. Therefore, concentration-dependent experimental results may be important in efficacy experiments such as anti-inflammatory or cell proliferation, but are omitted in this study. In particular, in cell proliferation experiments, it is important to compare the differences in cell proliferation rates of each compound according to concentration and time rather than comparing a single concentration and a single time point. In Figure 6, the possibility that DE and KL also reduce cell numbers at high concentrations cannot be ruled out.

We thank the Reviewer for this important comment. The aim of this pivotal study was first to compare the anti-inflammatory activity of the novel compounds with well-defined GCs such as DEX. We chose the dose for DEX based on data in the literature, and we used 1µM as a starting point to investigate the potency of the compounds and their potential side effects. In fact, this hormone concentration has been shown to saturate the receptor and has a potent transcriptional effect on keratinocytes. We agree with the Reviewer that each molecule may have different optimal concentrations for efficacy or for binding to GR, but we trust that our work could be the basis for a consecutive study on an experimental model more reliable to reproduce in vivo effects, such as 3D-skin equivalents. Also in fibroblasts proliferation experiments, we firstly aimed to make a comparison between DEX and the novel compounds. We did not investigate higher doses and times, as we already saw significant differences at the dose and time that we chose based on literature data for known GCs. We have modified the Introduction section to better focus on the aim of this pivotal study (page 2), and added some considerations in the Discussion (pages 13-14) and Study limitations sections (page 19).